# Altered Carbohydrate Allocation Due to Soil Water Deficit Affects Summertime Flowering in Meiwa Kumquat Trees

**Naoto Iwasaki \*, Asaki Tamura and Kyoka Hori**

School of Agriculture, Meiji University, Kawasaki, Kanagawa 214-8571, Japan; a.tamura1995@gmail.com (A.T.); kyohori66@gmail.com (K.H.)

\* Correspondence: iwasaki@meiji.ac.jp; Tel.: +81-44-934-7817

**Abstract:** The summertime flowers of the ever-flowering Meiwa kumquat (*Fortunella crassifolia* Swingle) are the most useful for fruit production in Japan; however, summertime flowers bloom in three or four successive waves at approximately 10 day intervals, resulting in fruit of different maturity occurring on the same tree. Soil water deficit (SWD) treatment has been shown to reduce the flowering frequency and improve harvest efficiency; therefore, in this study, the effects of SWD treatment on the accumulation of soluble sugars in each tree organ above-ground were examined and it was discussed how SWD affects the whole-tree water relations and sugar accumulation by osmoregulation. The number of first-flush summertime flowers was higher in SWD-treated trees than non-treated control (CONT) trees (177.0 and 58.0 flowers, respectively), whereas the second- and third-flush flowers were only observed in CONT trees. The soluble sugar content was higher in SWD treated trees than CONT trees for all organs and tended to be higher in current-year organs than previous-year organs; however, when the sugar content of the current-year spring stems exceeded approximately 100 mg g$^{-1}$ dry weight, the current-year leaf water potential decreased sharply and the rate of increase in the number of first-flush flowers also tended to decrease. SWD treatment significantly increased the total sugar content of the xylem tissue of the scaffold branches to three times the value in CONT trees ($p = 0.001$); however, the increase was observed even in sucrose, a disaccharide, similar to that in monosaccharides such as glucose and fructose. These results suggest that the increased sugar levels in the xylem tissue resulted from not only osmoregulation but also other factors as well; therefore, these sugars may affect whole-tree water relations as well as the development of flower buds.

**Keywords:** osmoregulation; sugar accumulation; water stress; xylem tissue

## 1. Introduction

Meiwa kumquat (*Fortunella crassifolia* Swingle) is an ever-flowering tree that usually blooms in spring (April), summer (June to August), and autumn (September) in Japan. The spring flowers appear on the previous year shoots, but are very few in number and not usually used for fruit production. By contrast, the summertime flowers appear on each axil of the current spring shoots that emerge around May and are the most useful for fruit production [1]; therefore, the flower bud differentiation period of the summertime flowers is regarded as approximately 1 month from the termination of spring shoot elongation to flowering [2]; however, the flowering of summertime flowers is divided into three to four separate time points, and when flowers do not bear fruitlets, new flowers appear at the same axil after 10–14 days. The resulting flowers are known as the first-, second-, and third-flush flowers in Japanese production areas [1,3]. The fruits from first-flush flowers tend to grow well and

mature early. The fruit quality also tends to be good in first-flush flowers. Since there are few first-flush flowers—the number of first-flush flowers also varies annually—the second-flush flowers are used for fruit production in general [1].

Many previous studies on Meiwa kumquat have shown that the application of a soil water deficit (SWD) for 2–4 weeks during the flower bud differentiation period increases the number of first-flush flowers [4–11]; however, the optimal duration of SWD treatment and its effectiveness varies between years. Iwasaki et al. [7] found that severe drought stress that reduced the predawn leaf water potential to below −1.7 MPa or increased the leaf abscisic acid (ABA) content to above 5 nmol $g^{-1}$ dry weight (DW) did not increase the number of first-flush flowers, suggesting that the leaf water potential needs to be maintained between 0 and −1.7 MPa; however, this condition is difficult to achieve because the leaf water potential fluctuates not only with soil water content but also with the relative humidity of the atmosphere [6,11].

Iwasaki and Hiratsuka [8] reported that the leaf water potential tends to be lower in Meiwa kumquat trees with larger trunk and scaffold branches under drought conditions. They also noted that transpiration from the trunk surface does not change despite rapid decreases in leaf transpiration under drought conditions, resulting in continuous water loss from the trunk at approximately one-sixth of the leaf transpiration rate. In some woody species, water stored in the trunk is transported to organs such as the leaves when required, but this process does not occur in isohydric plants [12–14]. Isohydric plants maintain a constant midday leaf water potential under both abundant water and drought conditions by reducing stomatal conductance to limit transpiration when required [14,15]. Thus, the kumquat tree is considered isohydric because SDW treatment rapidly decreases the transpiration rate in the leaves [9], and the water stored in the trunk does not migrate to the leaves. Plants that have encountered drought actively accumulate soluble sugars and free amino acids to prevent cell dehydration, via a process known as osmoregulation [16]. Thus the migration of water between the organs within a tree may be related to the amount of soluble sugars that have accumulated in that organ.

In the present study, the accumulation of soluble sugars in the organs of Meiwa kumquat trees under SWD treatment was investigated and considered the relationship between the sugar content of each organ and the number of first-flush flowers.

## 2. Materials and Methods

### 2.1. Plant Materials and Treatment

Six eighteen-year-old Meiwa kumquat trees that had been grafted onto trifoliate orange [*Poncirus trifiliata* (L.) Raf.] were used in the experiment, which was carried out during the growing season in 2016. The trees were planted in 29-L non-woven fabric pots containing commercially available humus-rich soil and each tree had a height and canopy diameter of approximately 1.2 and 1.5 m, respectively. Bud burst and elongation of spring shoots began around 25 April 2016, and were completed around 20 May 2016.

Three trees were allocated to each of the SWD treatment and non-treated control (CONT) groups on June 4 and maintained for 12 days in a greenhouse at Meiji University (Kawasaki, Kanagawa, Japan; 35°61′ N, 139°55′ E). For the SWD treatment, the soil water content was reduced by withholding irrigation, and a small volume of water (approximately 300 mL) was applied when the soil water content fell to <15%. During the treatment, the soil water content was monitored at a depth of approximately 10 cm between the trunk and the pot rim using an $ECH_2O$ soil moisture sensor (EC-5) with an Em50 digital data logger (METER Group, Inc., Pullman, WA, USA). The leaf water potential was determined on June 8, 11, and 14 using a psychrometer (WP4-T; METER Group, Inc., Pullman, WA, USA) to monitor the severity of water stress in the trees. One matured leaf in the middle of two current shoots per tree was collected at 9:00 a.m. on each measurement day.

### 2.2. Observation of Flowering Behavior and Determination of Total Soluble Sugars and Starch

Ten current-year spring shoots of approximately 10 cm length were selected from each tree to observe flowering behavior. The number of flowers that opened on each shoot was recorded each day from 20 June to 31 July, and the number of first-flush flowers and the total number of flowers per 10 shoots per tree was counted. One scaffold branch per tree was harvested on 15 July at the end of SWD treatment, and the plant parts were categorized into current-year spring leaves (current leaves), current-year spring stems (current stems), previous-year leaves (previous leaves), previous-year stems (previous stems), and scaffold branch, which included branches >2 years old. The scaffold branch was then further divided into bark and wood (xylem tissue). The fresh weight of each organ was measured, following which the organs were stored at −70 °C for further analysis. To analyze the soluble sugar and starch contents of the organs, the samples were freeze-dried and ground to powder. Soluble sugars were then extracted by adding 0.1 g of dried sample to 50 mL of 80% ethanol and leaving for 24 h at room temperature, following which the ethanol was removed from the extract using a rotary evaporator (N-1000; Tokyo Rika Kikai Co., Tokyo, Japan) at 35 °C and the remaining aqueous phase was diluted to 50 mL. The total sugar content in 1 mL of the diluted sample was then determined using the anthrone method. The starch content of the organs was determined with the starch-iodine colorimetric method [17] using the alcohol-insoluble solids (AIS) that were obtained during sugar extraction. The soluble sugar and starch contents of each plant part were expressed as mg $g^{-1}$ dry weight (DW). The water content (WC) of each plant part was calculated as WC = (FW − DW)/FW where FW is the fresh weight.

### 2.3. Sugar Composition of the Xylem Tissue and Bark Extracts of the Scaffold Branch

Following the extraction of sugars from each dried sample as described above, the aqueous phase that remained after ethanol removal was lyophilized. The residue was then dissolved in 2 mL of ultrapure water to prepare an analytical sample, which was filtered using a 0.45 μm membrane filter before being injected into a high-performance liquid chromatography (HPLC) system (LC-2000Plus series, JASCO, Tokyo, Japan). Sugars were separated in a Shodex SC1011 column (Shodex SC1011; Showa Denko K.K., Tokyo) that was maintained at 80 °C using ultrapure water as the mobile phase with a flow rate of 1.0 mL $min^{-1}$. Sucrose, glucose, and fructose were then identified and quantified by comparison with peaks of known standards using a refractive index detector (RI-2031; JASCO, Tokyo, Japan).

### 2.4. Statistical Analyses

Each treatment was performed on three trees (one tree per plot) and differences between the treatments were assessed using *t*-tests. The differences in sugar and starch content among the organs were compared using the Tukey HSD test; the coefficients of correlation and determination were obtained by regression analysis. Percentage values were arcsine transformed before analysis. Differences between the means were analyzed using the software KaleidaGraph v.4.5 (Synergy Software, Reading, PA, USA) when ANOVA was significant at *p* = 0.05 or *p* = 0.01, whereas BellCurve for Excel, v.2.00 (Social Survey Research Information Co., Ltd. Tokyo, Japan) was used for regression analysis.

## 3. Results

### 3.1. Changes in the Soil Water Content and Leaf Water Potential during SWD Treatment

The mean soil water content gradually decreased from 32.7% on June 8 to 12.1% on June 14 in the SWD treatment group but remained relatively constant in the CONT group (Table 1). The average leaf water potential decreased from −1.9 MPa on June 8 to −2.7 MPa on June 14 in the SWD-treated trees but remained at −1.6 to −1.7 MPa in the CONT trees.

**Table 1.** Changes in the soil water content and the leaf water potential at mid-morning during soil water deficit (SWD) treatment.

|  | 8 Jun | 11 Jun | 14 Jun |
|---|---|---|---|
| Soil water content (%) | | | |
| CONT | 41.6 [z] | 39.2 | 45.1 |
| SWD | 32.7 | 9.3 | 12.1 |
| Significance | ns [y] | * | * |
| Leaf water potential (Mpa) | | | |
| CONT | −1.75 | −1.61 | −1.62 |
| SWD | −1.92 | −2.38 | −2.73 |
| Significance | ns | * | * |

[z] Measurements were undertaken at 9:00 AM and values are mean of 3 replications. [y] ns and * indicate a non-significant and significant differences between the treatments at *p* = 0.05 by *t*-test, respectively.

### 3.2. Effect of SWD Treatment on Flowering Behavior

Blooming of the first- to fourth-flush flowers was observed by the end of July on the CONT trees, whereas only first-flush flowers were observed on the SWD-treated trees (Figure 1). The SWD-treated trees had a higher maximum number of flowers open per day than the CONT trees (46 and 26 flowers, respectively) and a significantly higher number of first-flush flowers (177.0 and 58.0 flowers per 10 shoots, respectively) (Table 2). The SWD-treated trees also had a higher total number of flowers than the CONT trees, though this difference was not significant.

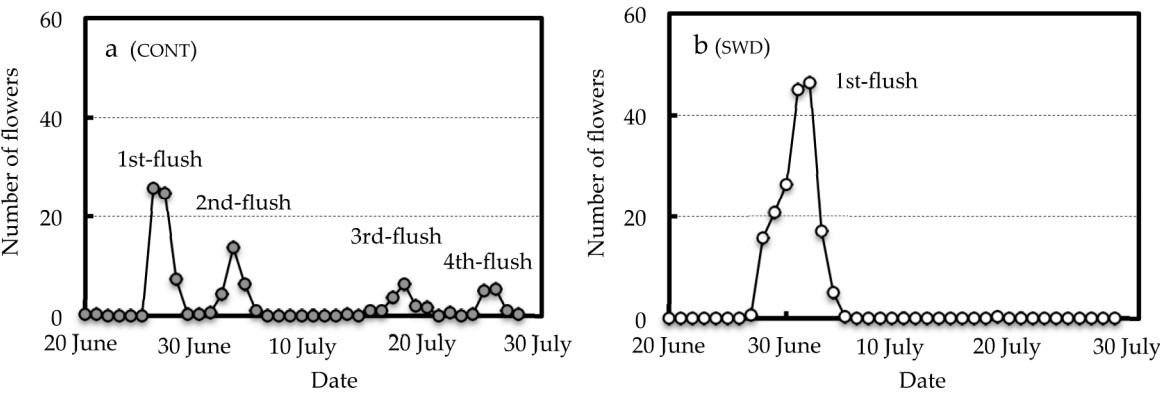

**Figure 1.** Difference of flowering behavior between control (CONT) (**a**) and soil water deficit (SWD) (**b**) in Meiwa kumquat trees. The number of flowers means the number of flowers that bloomed per day in 10 shoots per tree. Each plot indicates the mean of three replications.

**Table 2.** Effects of SWD treatment on the number of first-flush flowers and total flowers.

|  | 1st-Flush Flowers [z] | Total Flowers |
|---|---|---|
| CONT | 58.0 [y] | 113.7 |
| SWD | 177.0 | 177.3 |
| Significance | * [x] | ns |

[z] Flowers bloomed from June 26 to June 29 in CONT, June 27 to July 5 in SWD. [y] Number of flowers appeared 10 shoots per tree (*n* = 3). [x] * indicate significant difference between treatments at *p* = 0.05 by *t*-test.

### 3.3. Effects of SWD Treatment on the Soluble Sugar, Starch and Water Contents of the Plant Parts

For all plant parts, the soluble sugar content was significantly higher in the SWD-treated trees than in the CONT trees (Figure 2). In CONT trees, the soluble sugar content was significantly higher in the current leaves and stems than in the previous leaves and stems respectively. The xylem tissue of the scaffold branches showed the lowest soluble sugar content among all of the plant parts in the CONT

trees but also showed the greatest increase following SWD treatment (3.07 fold increase). By contrast, the bark of the scaffold branches had a higher soluble sugar content but experienced the lowest rate of increase following SWD treatment (1.16 fold increase). The starch content of the organs tended to be lower in the SWD treated trees, compared with the CONT trees, except for the current and previous leaves; however, this difference was only significant for the current stems. In the CONT trees, the starch contents were higher in the previous stems and leaves than in the current stems and leaves, respectively. There was no significant difference in water content between the SWD-treated trees and the CONT trees in any of the plant parts except for the current stems and the bark of the scaffold branches. In the CONT trees, the water content was highest in the current leaves (approximately 73%) and lowest in the xylem tissue of the scaffold branches (approximately 36%).

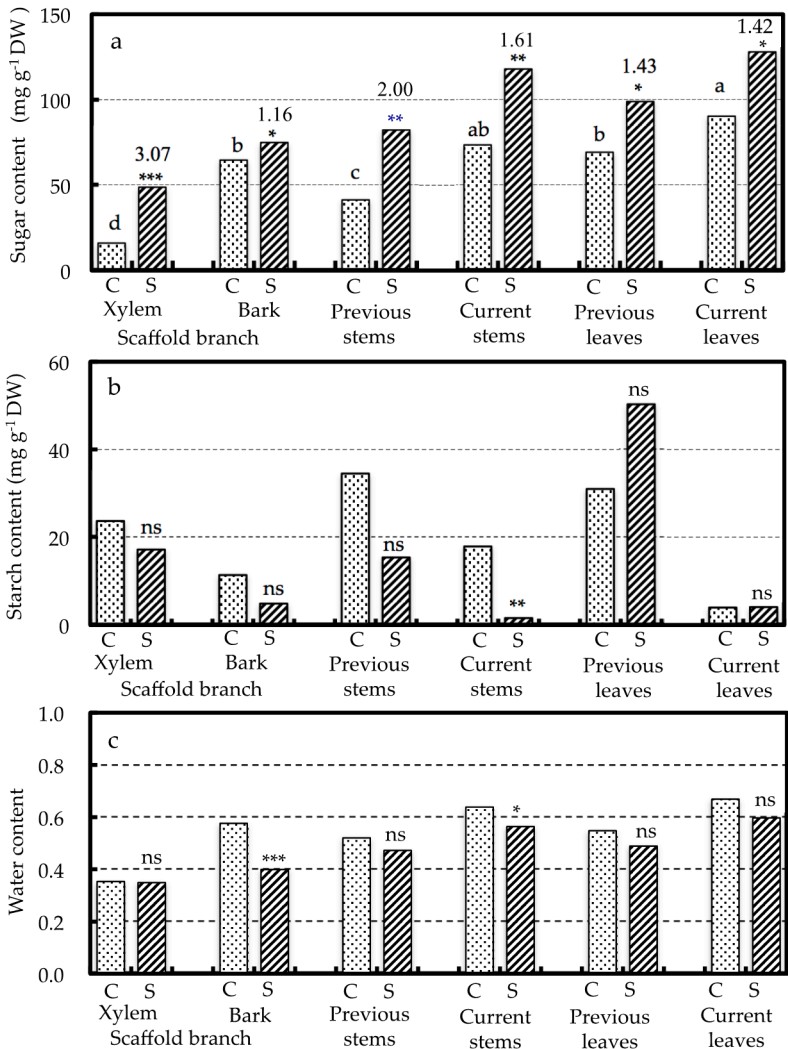

**Figure 2.** Soluble sugar (**a**), starch (**b**) and water (**c**) contents of various plant parts in Meiwa kumquat trees as affected by SWD treatment. C and S indicate CONT and SWD, respectively. The numbers above the bar of sugar content are the ratio of SWD to CONT. *, **, *** and ns indicate significant difference between the treatments at $p = 0.05$, 0.01 and 0.001 and non-significant by *t*-test, respectively ($n = 3$). Different letters above CONT bar indicate significant differences among the plant parts at $p = 0.05$ by Tukey HSD test ($n = 3$). Water content is expressed as the ratio of water to fresh weight.

Sucrose, glucose, and fructose were detected in both the xylem tissue and the bark of the scaffold branches (Table 3); however, sucrose was the predominant sugar in both tissues, whereas glucose was only detected at low levels in the bark. The concentrations of all three sugars were significantly higher in the xylem tissue of the SWD-treated trees than in that of the CONT trees, with the SWD treated trees exhibiting a four fold increase in sucrose and an approximately six fold increase in the monosaccharides. A similar pattern was also observed in the bark, although none of these differences were significant.

**Table 3.** Effect of SWD treatment on sugar compositions of xylem and bark in scaffold branch of Meiwa kumquat.

|  | Sucrose | Glucose | Fructose | Total |
|---|---|---|---|---|
| Xylem tissue |  |  |  |  |
| CONT | 3.4 [z] | 1.4 | 1.7 | 6.5 |
| SWD | 13.5 (4.0) [y] | 9.6 (6.6) | 9.8 (5.7) | 32.8 (5.0) |
| significance | * [x] | ** | ** | ** |
| Bark |  |  |  |  |
| CONT | 21.0 | 0.5 | 1.7 | 20.4 |
| SWD | 25.1 (1.2) | 1.1 (2.1) | 2.0 (1.2) | 28.3 (1.4) |
| significance | ns | ns | ns | ns |

[z] Values are the mean of 3 replications and expressed by mg g$^{-1}$ DW. [y] Number in parentheses indicate the ratio of SWD to CONT. [x] * and ** indicate significant difference between treatments at $p = 0.05$ and $p = 0.01$, respectively, by *t*-test.

### 3.4. Relationships between Sugar Content of the Current Spring Stems and the Number of First-Flush Flowers or Leaf Water Potential of the Current Spring Leaves

The number of first-flush flowers was positively correlated with the sugar content in each tree organ, but significantly negatively correlated with the water potential of the current leaves (Table 4). For both relationships, the correlation coefficients were highest in the current stems ($p = 0.01$), however, when the sugar content of the current stems exceeded approximately 100 mg g$^{-1}$ DW, the current leaf water potential rapidly decreased while the number of first-flush flowers plateaued (Figure 3).

**Table 4.** Correlation coefficients between sugar contents in each tree organ and the number of first-flush flowers or the current leaf water potential.

|  | Number of 1st-Flush Flowers | Current Leaf Water Potential |
|---|---|---|
| Scaffold branch |  |  |
| Xylem | 0.8385 * [z] | −0.8727 * [z] |
| Bark | 0.8395 * | −0.9011 * |
| Stems |  |  |
| Previous year stems | 0.8662 * | −0.9565 ** |
| Current-year spring stems | 0.9584 ** | −0.9567 ** |
| Leaves |  |  |
| Previous year leaves | 0.7015 | −0.9400 ** |
| Current-year spring leaves | 0.7055 | −0.7649 |

[z] ** and * indicate significant at $p = 0.01$ and $p = 0.05$, respectively.

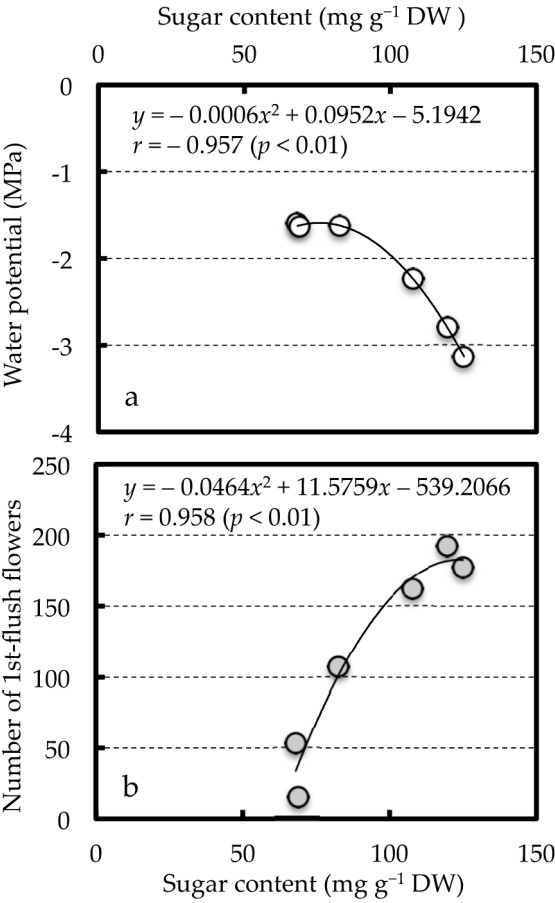

**Figure 3.** Relations between sugar content of the current shoot and the water potential of current leaves (**a**) or the number of first-flush flowers (**b**). The number of first-flush flowers indicates the number of flowers that opened in 10 current spring shoots per tree.

## 4. Discussion

In this study, it was found that SWD treatment reduced the frequency of sequential blooming of summertime flowers and increased the number of first-flush summertime flowers in Meiwa kumquat; however, the total number of flowers did not significantly change, indicating that SWD promoted flower bud development rather than differentiation. These results support the findings of previous studies [4,6] and suggest that the application of SWD treatment approximately two weeks after the cessation of spring shoot elongation during flower bud differentiation would accomplish uniform fruit maturation, removing the need to evaluate fruit maturity at harvest and thus simplifying harvesting practices.

In many crops, deficit irrigation is known to increase fruit sugar content without decreasing the yield [18–21], and nowadays, the water stress is often applied to produce high-quality fruits. We found that kumquat trees accumulated considerable amounts of sugars in their scaffold branches, particularly in the xylem tissue, under SWD conditions. Although the scaffold branches had a lower total soluble sugar content per gram dry weight than the leaves, they exhibited a greater proportional increase following SWD treatment compared with the CONT trees (3.1 fold increase in xylem tissue while a 1.4 fold increase in the leaves). In this study, the trunk was divided into bark and wood. The wood is the xylem tissue consists of vessels, tracheid, parenchymal cell, etc., but since the vessels and tracheid are dead cells, sugars cannot be actively absorbed into the cells [22,23]; therefore, we concluded that most sugars are accumulated in the xylem parenchyma cells. In the xylem tissue, sucrose was detected alongside glucose and fructose, and the sucrose content also increased alongside glucose or fructose under SWD conditions. Since the water content of the xylem tissue did not change as a result of SWD treatment, the increase in sugar content might involve not only osmoregulation [16,24] but also other

factors. Trees growing in cold areas such as birch are known to accumulate sucrose, raffinose, and stachyose in xylem parenchyma cells during the cold season [25]. These sugars are considered to not only increase the freeze-tolerance of cells but have a stabilizing effect of cell membrane under stress conditions such as freezing, desiccation and high temperature, and that the effect is higher for trisaccharides or disaccharides having a higher molecular weight than monosaccharides [26].

It has been previously shown that the carbohydrate contents of the leaves or bark of shoots that produce the next flowers gradually increase during flower bud differentiation/development period in Satsuma mandarin (*Citrus unshiu*) trees growing outdoors in Valencia, Spain [27], or in early-heating plastic houses in Fukuoka, Japan [28], although they sometimes decrease temporarily; however, none of these studies found a direct relationship between flower bud differentiation and the sugar or starch content. In the present study, the number of first-flush flowers was most closely related to the sugar content of the current shoots that set flowers for fruit production. Similarly, Niii and Okamoto [29] reported that the amount of stored carbohydrates influenced the development of flower buds in Satsuma mandarin, with flower buds showing better development when old leaves existed during budburst. It has also been shown that buds of peach (*Prunus persica*) actively absorbed carbohydrates during dormancy release, which are used for growth metabolism and thereby induce bud development [30]. In addition, Nakajima et al. [31] reported that water stress from early September through December increased the number of 'Tosa Buntan' pomelo (*Citrus grandis*) flowers and caused them to bloom earlier the following spring. Consequently, the authors considered that water-stressed trees were forced into dormancy to promote flower induction, and released from dormancy by water stress in a similar way to the flower buds of coffee (*Coffea arabica*) [32]. SWD treatment may have caused flower bud dormancy to occur earlier in kumquat, although the dormant period, occurring immediately after the cessation of spring shoot elongation, may be short. In recent years, sugars such as sucrose have been reported to be important signals that regulate bud outgrowth and act before hormones in releasing apical dominance [33,34], i.e., sucrose promotes auxin export from the bud, then the axillary buds have sustained outgrowth. Since the flower buds of kumquat are formed in the axils of shoots, the mechanism involved in their development may be similar to that of apical dominance; therefore, the sucrose accumulated in xylem, just as in glucose and fructose, may have promoted the development or budburst of floral buds that originally should have been the second- or third-flush flowers, and made them the first-flush flowers.

Consequently, such sugar accumulation may decrease the water potential of the xylem tissue, resulting in the active accumulation of water due to osmoregulation. Although the sugar content of the trunk was not investigated in this experiment, the sugar content of the scaffold branches is expected to reflect this. Both the trunk and scaffold branches can be regarded as water reservoirs in many tree species [12,35,36]. Nevertheless, the rapid decrease in leaf water potential that was observed during SWD treatment in this experiment indicated that the water supply from the trunk and scaffold branches to the leaves was restricted. Iwasaki et al. [9] reported that the water potential of the trunk xylem tissue was always lower than the leaf water potential in kumquat trees under SWD conditions, which may result from the large accumulation of sugars. It was also found that the relationship between the number of first-flush flowers or current leaf water potential and the sugar content of the current stems changed when the sugar content reached approximately 100 mg g$^{-1}$ DW. Similarly Iwasaki et al. [7] reported that severe water stress expressed by a leaf water potential below −1.7 MPa or a leaf ABA content above 5 nmol g$^{-1}$ did not increase the number of first-flush flowers. Thus, factors related to the level of water stress, such as leaf water potential and sugar content of the plant tissues, appear to have a limited range of influence on the number of first-flush flowers. Although it is necessary to investigate the detailed mechanism of sugar accumulation under drought conditions, the xylem tissue of the trunk and scaffold branches seems to play an important role in the whole-tree water relations through active sugar accumulation.

**Author Contributions:** Conceived and designed the experiments: N.I.; performed the experiments: A.T. and K.H.; analyzed the data: A.T. and N.I.; wrote the manuscript: A.T. and N.I. All authors have read and agreed to the published version of the manuscript.

**Funding:** This research received no external funding.

**Conflicts of Interest:** The authors declare no conflict of interest.

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
