# Peer review of "Altered Carbohydrate Allocation Due to Soil Water Deficit Affects Summertime Flowering in Meiwa Kumquat Trees"

_horticulturae, doi:10.3390/horticulturae6030049_

Round 1
Reviewer 1 Report
The manuscript ''Soil water deficit treatment aimed to increase the number of first-flush summertime flowers affects the sugar content in the various parts of Meiwa kumquat (Fortunella crassifolia Swingle) trees'' by Naoto Iwasaki et al., is well written and the findings are important to better understand the response of fruit trees to water stress and thus to improve theirs production.
However, before the publication, it needs to be improved. Following the specific comments:
Title: It is too long, I suggest to shorten it.
L-20: It is the only case where there is a hyphen in ''SWD-treated'', remove it.
Introduction: The introduction must be expanded. Give more insights into the state of the art.
L-32: Add here the name in Latin of the tree species, then use only the common name.
L-45: I would say ''However, this condition is difficult...'' instead of ''However, this is difficult...''.
L67-69: How many trees did you consider? The experiment was carried out in 2016, throughout the whole year?
L-79: How did you choose the leaves to measure? Did you measure always the same two?
L-113: Specify the plot.
L-114: Retype the sentence ''And differences in sugar...'', don't start the sentence with And.
L274: format the caption.
L338: The paragraph regarding the conclusion is missing. I suggest to add it to emphasize the importance of your finding.
References: Check again all the references. Journal titles are sometimes abbreviated, sometimes not. I suggest adding more references. Widen the state of the art in the introduction.
Author Response
Dear Reviewer 1.
Thank you very much for the careful review and the useful comments. We would like to explain the response to the comments as follows.
1.Comments and Suggestions for Authors
Title: It is too long, I suggest to shorten it.
Answer: I tried shortening it, but think a little more.
L-20: It is the only case where there is a hyphen in ''SWD-treated'', remove it.
Answer: Changed according to advice.
Introduction: The introduction must be expanded. Give more insights into the state of the art.
L-32: Add here the name in Latin of the tree species, then use only the common name.
Answer: Changed according to advice.
L-45: I would say ''However, this condition is difficult...'' instead of ''However, this is difficult...''.
Answer: Changed according to advice.
L67-69: How many trees did you consider? The experiment was carried out in 2016, throughout the whole year?
Answer: Changed as line 67 - 73.
L-79: How did you choose the leaves to measure? Did you measure always the same two?
Answer: Changed as line 81 - 82.
L-113: Specify the plot.
Answer: I'm sorry, but I don't know where the reviewer is pointing.
L-114: Retype the sentence ''And differences in sugar...'', don't start the sentence with And.
Answer: Changed according to advice.
L274: format the caption.
Answer: The format was changed.
L338: The paragraph regarding the conclusion is missing. I suggest to add it to emphasize the importance of your finding.
Answer: Changed according to advice. Line 340 - 352.
References: Check again all the references. Journal titles are sometimes abbreviated, sometimes not. I suggest adding more references. Widen the state of the art in the introduction.
Answer: I'm sorry, but I didn't have enough time.
Sincerely yours,
Authors
Reviewer 2 Report
The authors need to be clear that the changes in sugar & water content in the “xylem” that is presented and discussed in this manuscript is in branch tissues enriched with xylem vessels.
Page 1, line 15-17. In this sentence, the authors state that they measure sugars “in each tree organ”, when sugar measurements were only performed in leaves, stems and branches. In addition to measuring soluble sugars the authors also measured starch.
Page 1, line 21. Change organs to scions
Page 1, lines 26-28. The authors need to explain why is an increase in sucrose is significant, as a reader not familiar with the effects of SWD on sugar accumulation will be confused.
Page 1, lines 27-28. The concluding sentence leaves the reader wondering what the focus of the paper is. The title would suggest that manuscript address the relationship between sugar content and the pattern of flowering in response to SWD. However, the concluding paragraph is about sugar levels and osmomoregulation (+ “other factors”) in the xylem.
Page 1, line 33. It would be more informative to state that spring flowers have a low propensity to set fruit.
Page 1, line 39. Delete “Many” as only three publications are referenced.
Page 2, line 70. Change “sprouting” to “bud burst”
Page 2, Plant materials and treatment. Its not clear from this section when the SWD treatment was applied and for how long.
Page 3, line 13-14. “Each treatment” and “treatments” implies that multiple SWD experiments were performed.
Page 4, line 35. Information missing from this sentence. When was the first flush initiated in the CONT trees and what was the duration of the first flush? How many flushes occurred in CONT trees, and when did the fourth flush end? What was the number of flowers produced in each of the flush? For the SWD trees, when was the flush initiated and what was the duration? How many flushes were initiated? How many flowers were produced for the single flush? Comparison of total flowers for CONT and SWD.
Page 5. Line 56. State the plant parts.
Page 7, lines 274-278. The authors need to rewrite this figure legend. It is confusing to read with a period at the end of each line.
Page 8, 299-301. The authors speculate that increased soluble sugars in the stem may have stimulated bud burst of floral buds. This hypothesis is in agreement with sugars being attributed to branching and axillary bud outgrowth (Barbier et al., 2019). It would be good to include a sentence about the role of sugars in bud release.
Page 7, lines 280-287. SWD results in a continuous flush of flowers. Is there any change in the number of fruit that set between the SWD and CONT?
Page 8, 299-301. It would be nice if the authors elaborated the point discussed in the sentence, by including a supporting statement that sugars mediate bud release in decapitated stems (see reviews: Barbier et al., 2015; and Barbier et al., 2019)
Author Response
Dear Reviewer 2.
Thank you very much for the careful review and the useful comments. We would like to explain the response to the comments as follows.
- Comments and Suggestions for Authors
The authors need to be clear that the changes in sugar & water content in the “xylem” that is presented and discussed in this manuscript is in branch tissues enriched with xylem vessels.
Answer: When we use "xylem", we don't mean the "xylem vessel", but the "tissue" that contains the vessels and parenchyma. Therefore, we changed to "xylem tissue" instead of "xylem".
Page 1, line 15-17. In this sentence, the authors state that they measure sugars “in each tree organ”, when sugar measurements were only performed in leaves, stems and branches. In addition to measuring soluble sugars the authors also measured starch.
Answer: The word "above ground" was added.
Page 1, line 21. Change organs to scions
Answer: Since scion includes all parts except rootstock, it does not refer to branches or leaves separately, so we did not change the organs.
Page 1, lines 26-28. The authors need to explain why is an increase in sucrose is significant, as a reader not familiar with the effects of SWD on sugar accumulation will be confused. 
Answer: The abstract has a limited number of words, so this is described in the discussion.
Page 1, lines 27-28. The concluding sentence leaves the reader wondering what the focus of the paper is. The title would suggest that manuscript address the relationship between sugar content and the pattern of flowering in response to SWD. However, the concluding paragraph is about sugar levels and osmomoregulation (+ “other factors”) in the xylem.
Answer: In this study, we investigated not only the effects of SWD on the number of flowers, but also the effects of SWD on the sugar content of various parts of the tree.
Page 1, line 33. It would be more informative to state that spring flowers have a low propensity to set fruit.
Page 1, line 39. Delete “Many” as only three publications are referenced.
Answer: added the references.
Page 2, line 70. Change “sprouting” to “bud burst”
Answer: It has changed.
Page 2, Plant materials and treatment. Its not clear from this section when the SWD treatment was applied and for how long.
Answer: It is described in Line 74.
Page 3, line 13-14. “Each treatment” and “treatments” implies that multiple SWD experiments were performed.  
Answer: In this experiment, SWD and CONT are included.
Page 4, line 35. Information missing from this sentence. When was the first flush initiated in the CONT trees and what was the duration of the first flush? How many flushes occurred in CONT trees, and when did the fourth flush end? What was the number of flowers produced in each of the flush? For the SWD trees, when was the flush initiated and what was the duration? How many flushes were initiated? How many flowers were produced for the single flush? Comparison of total flowers for CONT and SWD.
Answer: It is difficult to explain because it varies from year to year. You can see from Figure 1.
Page 5. Line 56. State the plant parts.
Answer: I'm sorry, but I don't know where the reviewer is pointing.
Page 7, lines 274-278. The authors need to rewrite this figure legend. It is confusing to read with a period at the end of each line.
Answer: The format was changed.
Page 8, 299-301. The authors speculate that increased soluble sugars in the stem may have stimulated bud burst of floral buds. This hypothesis is in agreement with sugars being attributed to branching and axillary bud outgrowth (Barbier et al., 2019). It would be good to include a sentence about the role of sugars in bud release.
Answer: I'm sorry, but I don't know where the reviewer is pointing.
Page 7, lines 280-287. SWD results in a continuous flush of flowers. Is there any change in the number of fruit that set between the SWD and CONT?
Answer: I'm sorry, but I don't know where the reviewer is pointing.
Page 8, 299-301. It would be nice if the authors elaborated the point discussed in the sentence, by including a supporting statement that sugars mediate bud release in decapitated stems (see reviews: Barbier et al., 2015; and Barbier et al., 2019)
Answer: We could not find the paper by Barbier et al. Please let me know the details.
Sincerely yours,
Authors
Round 2
Reviewer 2 Report
Page 7, lines 280-287. SWD results in a continuous flush of flowers. Is there any change in the number of fruit that set between the SWD and CONT?
The authors indicate that Meiwa kumquat initiates, four flushes of reproductive growth, starting in the spring (page 1, lines 32-38). Further, flowers produced in the spring are “not used in fruit production”, while flowers produced in the later flushes are “the most useful for fruit production”. Are the authors stating that fruit set is low in the first flush and high in the later three summer? If SWD causes trees to produce a single flush in the spring, is there a change in the total number of fruits that set between SWD and CONT?
Page 8, 299-301. It would be nice if the authors elaborated the point discussed in the sentence, by including a supporting statement that sugars mediate bud release in decapitated stems (see reviews: Barbier et al., 2015; and Barbier et al., 2019)
Barbier et al., (2015) Ready, steady, go! A sugar hit starts the race to shoot branching. Current opinion in Plant Biology. 25:39-45
Barbier et al., (2019) An update on the signals controlling shoot branching. Trends in Plant Science. 24:220-236.
Author Response
Page 7, lines 280-287. SWD results in a continuous flush of flowers. Is there any change in the number of fruit that set between the SWD and CONT?
Reply: There is no difference in yield between SWD-treated and control trees. These results have been reported in previous papers(Iwasaki et al., 2000;Iwasaki and Yamaguchi, 2004). Since this paper is not a report on yield, it is not described in detail. However, Table 2 has been replaced, showing that there is no difference between the number of first-flush flowers in SWD and the total number of summertime flowers in cont.
The authors indicate that Meiwa kumquat initiates, four flushes of reproductive growth, starting in the spring (page 1, lines 32-38). Further, flowers produced in the spring are “not used in fruit production”, while flowers produced in the later flushes are “the most useful for fruit production”. Are the authors stating that fruit set is low in the first flush and high in the later three summer? If SWD causes trees to produce a single flush in the spring, is there a change in the total number of fruits that set between SWD and CONT?
Reply: Spring flowers, summer flowers and autumn flowers are not the first-, second- and third-flush flowers. The first-, second- and third-flush flowers are all summertime flowers. The summertime flowers appear on spring shoots, while spring flowers do on the previous year shoots which grew previous year season. Usually, the summer flowers of Meiwa Kumquat bloom several times on the same shoots that have sprouted around May of the year. Flowering of the 1st-, 2nd- and 3rd-flush flowers does not accompany the growth of new shoots. They bloom on the same spring shoot. The SWD-treatment makes the summer flowers that bloom several times into one time. The total number of summertime flowers in SWD treated trees is not different significantly from CONT trees.
Page 8, 299-301. It would be nice if the authors elaborated the point discussed in the sentence, by including a supporting statement that sugars mediate bud release in decapitated stems (see reviews: Barbier et al., 2015; and Barbier et al., 2019)
Reply: Added the sentence, line 336.